# TEMPERED ADVERSARIAL NETWORKS

**Mehdi S. M. Sajjadi & Giambattista Parascandolo & Arash Mehrjou & Bernhard Schölkopf**
Max Planck Institute for Intelligent Systems, Tübingen, Germany
{msajjadi,gparascandolo,amehrjou,bs}@tue.mpg.de

## ABSTRACT

Generative adversarial networks (GANs) have been shown to produce realistic samples from high-dimensional distributions, but training them is considered hard. A possible explanation for training instabilities is the inherent imbalance between the networks: While the discriminator is trained directly on both real and fake samples, the generator only has control over the fake samples it produces since the real data distribution is fixed by the choice of a given dataset. We propose a simple modification that gives the generator control over the real samples, leading to a tempered learning process for both generator and discriminator. The real data distribution passes through a *lens* before being revealed to the discriminator, balancing the training process by gradually revealing more detailed features necessary to produce high-quality results. The proposed module automatically adjusts the learning process to the current strength of the networks, yet is generic and easy to add to any GAN variant. In a number of experiments, we show that this is a promising technique to improve quality, stability and/or convergence speed across a range of different GAN architectures (DCGAN, LSGAN, WGAN-GP).

## 1 INTRODUCTION

The basic setup of Generative Adversarial Networks (GANs) (Goodfellow et al., 2014) is to train a generator $G$, which maps samples from random noise drawn from a distribution $\mathcal{Z}$ into samples of a fake distribution $G(\mathcal{Z})$ which are close in terms of some measure to a real world empirical data distribution $\mathcal{X}$. To achieve this goal, a discriminator $D$ is trained to provide feedback in the form of gradients for $G$. GANs are infamous for being difficult to train and sensitive to small changes in hyper-parameters (Goodfellow et al., 2016). While several GAN variants have been introduced to address the problems encountered during training, finding stable and more reliable training procedures is still an open research question (Lucic et al., 2017).

In this work we propose a general and dynamic, yet simple to implement extension to GANs that encourages a smoother training procedure. We introduce a *lens* module $L$ which gives the generator control over the real data distribution $\mathcal{X}$ before it enters the discriminator. By adding $L$ between the real data samples and $D$, we allow training to self-stabilize by automatically balancing a reconstruction loss with the current performance of $G$ and $D$. While the generator in a regular GAN chases a fixed distribution $\mathcal{X}$, the proposed lens moves the target distribution closer to the generated samples which leads to a better optimization behavior.

## 2 METHOD

In GANs, the generator maps a random distribution $\mathcal{Z}$ to $G(\mathcal{Z})$ which is in the same space as the real data distribution $\mathcal{X}$. While the discriminator sees both real samples from $\mathcal{X}$ and fake samples from $G(\mathcal{Z})$, the generator only has control over the samples it produces itself, *i.e.*, it has no control over $\mathcal{X}$ which is fixed throughout training. To resolve this asymmetry, we add a lens module $L$ which modifies the real data distribution $\mathcal{X}$ before it is passed to the discriminator. In practice, the lens is implemented as a neural network. The only change in the GAN architecture is consequently the input to the discriminator, which changes from $\{\mathcal{X}, G(\mathcal{Z})\}$ to $\{L(\mathcal{X}), G(\mathcal{Z})\}$ (see Appendix, Fig. 2, left).

We train the lens with two loss terms: an adversarial loss $\mathcal{L}_L^A$ and a reconstruction loss $\mathcal{L}_L^R$. The adversarial term maximizes the loss of the discriminator of the respective GAN architecture, *i.e.*, $\mathcal{L}_L^A \approx -\mathcal{L}_D$. For the original GAN variant, this leads to the following objectives:

$$\mathcal{L}_G = -\log(D(G(\mathcal{Z}))), \ \mathcal{L}_D = -\log(D(L(\mathcal{X}))) - \log(1 - D(G(\mathcal{Z}))), \ \mathcal{L}_L^A = -\log(1 - D(L(\mathcal{X})))$$

For the objectives for other GAN variants, see Appendix, Sec. A.1. A reconstruction term prevents the lens from converging to trivial solutions (*e.g.*, mapping all samples to zero):

$$\mathcal{L}_L^R = ||\mathcal{X} - L(\mathcal{X})||_2^2$$

The overall loss for the lens is $\mathcal{L}_L = \lambda \mathcal{L}_L^A + \mathcal{L}_L^R$. The lens automatically balances a good reconstruction of the original samples with the objective of mapping the real data distribution $\mathcal{X}$ close to the generated data distribution $G(\mathcal{Z})$ *w.r.t.* the probabilities given by $D$. As training progresses, the generated samples get closer to the real samples, *i.e.*, the lens can afford to reconstruct the real data samples better. Once the discriminator starts to see differences, the loss term $\mathcal{L}_L^A$ increases which makes $L$ shift the real data distribution $\mathcal{X}$ towards the generated samples, helping to keep $G(\mathcal{Z})$ and $L(\mathcal{X})$ closer together which yields better gradients during training. To make sure that $L$ eventually converges to the identity mapping, we smoothly decrease $\lambda$ from 1 to 0 (see Appendix, Sec. A.3).

## 3 EXPERIMENTS

Showing that modifications or additions to GANs lead to *better* results in any way is a delicate topic that has raised much controversy in the community. Most recently, the findings of Lucic et al. (2017) suggest that with a sufficient computational budget, any GAN architecture can be shown to perform at least as well or better than another. To avoid this fallacy, we follow common guidelines that are currently in use for training GANs and we conduct experiments with three different GAN frameworks: the original GAN formulation (Goodfellow et al., 2014), LSGAN (Mao et al., 2017) and WGAN-GP (Gulrajani et al., 2017). For the network architecture, we follow standard design patterns (see Appendix, Sec. A.2). In all experiments, the random weights for the initialization of the networks were identical for the GANs with and without a lens. All experiments have further been run with at least 3 different random seeds for the weight initialization to prevent chance from affecting the results. We test the lens module on the MNIST, Color MNIST, CelebA (Liu et al., 2015) and Cifar-10 datasets.

For evaluation, we report the *Fréchet Inception Distance* (FID) that has been shown to correlate better with the perceived image quality than the Inception score (Heusel et al., 2017). In our experimental section, we do not strive for state-of-the-art results, but rather we test how much of an effect the lens can have on the learning process. It turns out that the addition of a lens module can help improve results across various GAN frameworks. We hope that this insight will help ongoing efforts to understand and improve the training of GANs and other neural network architectures.

**Original GAN objective** Using the original GAN objective on MNIST with small networks, the best FID we could achieve was 42 while adding a lens leads to a much better FID of 22 (see Appendix, Sec. A.4.1 for more details). To have a highly multi-modal dataset that is still easily interpretable, a color MNIST variant has been proposed (Srivastava et al., 2017). Each sample is created by stacking three randomly drawn MNIST digits into the red, green and blue channels of an RGB image. As can be seen in Fig. 1 (top), the lens first scrambles the original dataset to make it look more similar to the generated samples. As the generator catches up, the lens slowly improves reconstruction. While the GAN produces decent results in the green color channel, it has collapsed in the red and blue channels, only achieving an FID of 63 (Fig. 1, bottom, first row). Adding the lens to the GAN stabilizes training and leads to much higher quality samples with an FID of 9. Visual inspection confirms that the GAN trained with a lens produces realistic results in all color channels. Note that the lens also makes the training procedure more robust against different random seeds, see Appendix, Fig. 5.

**LSGAN** We found the LSGAN variant to be sensitive to the random seed for the weight initialization of the networks. LSGAN without a lens on MNIST did not train in most cases, with the best run yielding FID scores of 19. With the lens, the networks always trained well, with the worst run producing FID scores of 16 and the best run giving FID scores of 14. On the Color MNIST dataset, we found LSGAN to perform similarly. The best run without a lens yielded FID scores of 90 and training stalled there due to starved gradients. Adding the lens made the networks produce meaningful results in all runs, producing FID scores between 14 and 22 from different random initializations.

On the CelebA dataset (Liu et al., 2015), LSGAN without a lens yielded a best FID score of 52 amongst all runs. Adding the lens helped the system stabilize and produce meaningful results in all runs, with best and worst FID scores between 32 and 37. The effects of the lens during training are shown in the Appendix, Fig. 8.

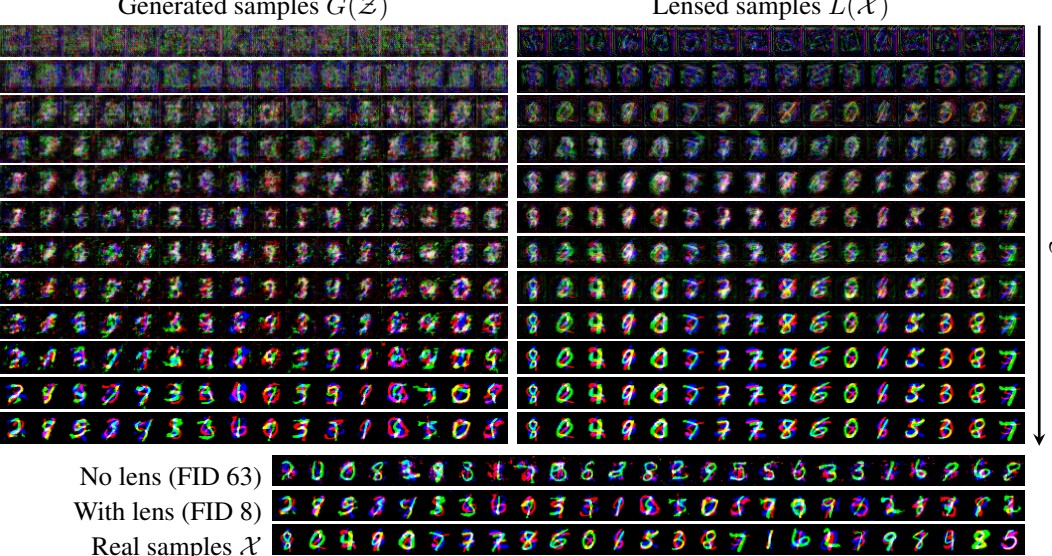

Figure 1: Results of DCGAN with a lens on the Color MNIST dataset (top). Reconstruction is gradually improved by $L$ as $G$ produces better samples. Once $L$ is a perfect identity function, $G$ adds remaining details and finally produces realistic results (FID 8, bottom, second row). In comparison, the GAN without a lens only manages to produce digits in the green color channel and produces noise in the other channels (FID 63, bottom, first row). Several runs with different random seeds for the weight initialization yielded similar results for both architectures, see Appendix, Fig. 5. Best viewed in color.

**WGAN-GP** Wasserstein GANs are generally believed to be more stable than other GAN variants, making it harder for tweaks to significantly improve sample quality. Nevertheless, our experiment on the Cifar-10 dataset with the same lens architecture showed that higher-quality results can be produced at an earlier training stage. The results are shown in the Appendix, Fig. 6. The model with a lens quickly surpasses the quality of the model without a lens and it takes some more training time for the GAN without a lens to catch up. When trained long enough, both models yield an FID of 39.

It is noteworthy that adding the lens can lead to faster training although $G$ and $D$ are initially trained on a data distribution $L(\mathcal{X})$ that is quite different from the real distribution $\mathcal{X}$ (see Appendix, Fig. 7). This result suggests that a tempered learning procedure can accelerate optimization of neural networks.

## 4 RELATED WORKS

Most closely related to our work, Arjovsky & Bottou (2017) and Mehrjou et al. (2017) add noise to the real samples or to real and fake samples with the idea of increasing the support of the distributions. The amount of noise is reduced manually during training. Our lens is not constrained in the mapping it can apply to balance the training procedure. Furthermore, the effect of the lens is automatically balanced with a reconstruction term that adjusts the intervention of the lens dynamically during training. A number of works yield better results by using multiple networks instead of one (Denton et al., 2015; Zhang et al., 2017). Such methods have the drawback that several GANs need to be trained which introduces a computational bottleneck. Most recently, Karras et al. (2018) produced convincing high-resolution images of faces by first learning the low-resolution images and then progressively growing both networks. However, this approach is limited to generating images.

## 5 CONCLUSION

We propose a generic module that leads to a dynamically self-adjusting progressive learning procedure of the target distribution in GANs. Whilst the method is simple, it may have significant potential as is highlighted in a number of experiments on several GAN variants. We hypothesize that similar modifications can be applied to improve optimization of other neural network architectures.

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

# A  APPENDIX

## A.1  OBJECTIVES FOR LSGAN AND WGAN-GP

In LSGAN (Mao et al., 2017), the log-loss is replaced by the squared distance. This leads to the adversarial loss

$$\mathcal{L}_G = ||D(G(\mathcal{Z})) - 1||_2^2 \tag{1}$$

for the generator, and

$$\mathcal{L}_D = ||D(G(\mathcal{Z}))||_2^2 + ||D(L(\mathcal{X})) - 1||_2^2 \tag{2}$$

for the discriminator. The lens works against the discriminator with the adversarial loss

$$\mathcal{L}_L^A = ||D(L(\mathcal{X}))||_2^2 \tag{3}$$

The discriminator or *critic* in the WGAN-GP variant (Gulrajani et al., 2017) outputs values that are unbounded, *i.e.*, there is no sigmoid activation at the after the last dense layer in Fig. 3. The objectives are

$$\mathcal{L}_G = -D(G(\mathcal{Z})) \tag{4}$$

for the generator, and

$$\mathcal{L}_D = D(G(\mathcal{Z})) - D(L(\mathcal{X})) \tag{5}$$

for the critic. Again, the lens works against the critic, so we use the adversarial objective

$$\mathcal{L}_L^A = D(L(\mathcal{X})) \tag{6}$$

for the lens for this GAN variant.

## A.2  ARCHITECTURE, TRAINING AND EVALUATION METRICS

Since it is desirable for the lens to turn into the identity mapping at some point during training, we have chosen a residual fully convolutional neural network architecture for the lens, see Fig. 3 (right). The network architecture and training procedure for the generator and discriminator depend on the chosen GAN framework. For the experiments with the original GAN loss and LSGAN, we use the DCGAN architecture along with its common tweaks (Radford et al., 2016), namely, strided convolutions instead of pooling layers, applying batch normalization in both networks, using ReLU in the generator and leaky ReLU in the discriminator, and Adam (Kingma & Ba, 2015) as the optimizer. See Fig. 3 (left) for an overview of the networks. For the WGAN-GP experiments, we used the implementation from Gulrajani (2017) which uses very similar models but the RMSProp optimizer (Hinton et al., 2012). We train the lens alongside the generator and discriminator and update it once per iteration regardless of the GAN variant. Note that the networks for the DCGAN and LSGAN experiments have intentionally been chosen not to have a very large number of feature channels to avoid memorization on small datasets which is why the results on an absolute scale are certainly not state of the art. We train using batch sizes of 32 and 64, a learning rate of $10^{-4}$ and we initialize the networks with the Xavier initialization (Glorot & Bengio, 2010).

For computational reasons, the FID scores are computed on sets of 4096 samples for the DCGAN and LSGAN experiments. While this is lower than the recommended 10k and should therefore not be compared directly with other publications, we found the sample size to be sufficient to capture relative improvements as long as sample sizes are identical. For the WGAN-GP experiments, we used sample sizes of 10k data points. The image size in all experiments is 32×32 pixels with 1 color channel for MNIST and 3 color channels for all other experiments.

## A.3  ADAPTATION OF $\lambda$ DURING TRAINING

To have a smooth transition from an adversarial lens to the identity mapping in $K$ steps, we adapt the value for $\lambda$ as

$$\lambda = \begin{cases} 1 - \sin(t\pi/2K), & t \leq K \\ 0, & t > K \end{cases} \tag{7}$$

for the $t$-th time step during training. The value of $\lambda$ over time can be seen in Fig. 2 (right). Once the lens converges to the identity mapping, training reduces to the original GAN architecture without

a lens. In all experiments, we set $K = 10^5$ unless specified otherwise. Lower values for $K$ lead to faster convergence, but to avoid introducing a new hyperparameter that needs to be tuned, and for simplicity, we choose the same value for all experiments. Note that this choice is clearly not optimal for all tasks and tuning the value can easily lead to even faster convergence and higher quality samples.

## A.4 EXPERIMENTS

### A.4.1 MNIST WITH ORIGINAL GAN OBJECTIVE

To analyze the behavior of the lens, we consider the case of fixed $\lambda = 1$, *i.e.*, the lens has no direct incentive to become perfect identity. Fig. 4 (top) shows generated and lensed samples at different training stages for this architecture. At the beginning of training, the lens scrambles the MNIST digits to look more similar to the generated images. As the generator catches up and produces digit-like samples, the lens can afford to improve reconstruction. Since the lens acts as a balancing factor between the $G$ and $D$, this leads to a very stable training procedure. However, even after 10M steps, the reconstruction of the lens still improves, as does the FID score of the generated samples (see FID plot in Fig. 4, bottom left). In comparison, the GAN without a lens converges much faster to better FID scores (Fig. 4, bottom right, green curve).

To accelerate the training procedure, we adapt the weight of $\lambda$ as explained in Sec. A.3. As this forces the lens to turn into a perfect identity mapping at some point during training, the process converges much more quickly and easily surpasses the quality of the GAN without a lens, yielding FID scores of 22 (with lens) vs. 42 (without lens).

### A.4.2 LSGAN OBJECTIVE

**MNIST** We found the LSGAN variant to be sensitive to the random seed for the weight initialization of the networks. LSGAN without a lens did not train in most cases, with the best run yielding FID scores of 19. With the lens, the networks always trained well, with the worst run producing FID scores of 16 and the best run giving FID scores of 14.

**Color MNIST** On the Color MNIST dataset, we found LSGAN to perform similarly. The best run without a lens yielded FID scores of 90 and training stalled there due to starved gradients. Adding the lens made the networks produce meaningful results in all runs, producing FID scores between 14 and 22 from different random initializations.

**CelebA** On the CelebA dataset, LSGAN was unstable, with a starving generator early on during training due to a perfect discriminator that did not provide gradients. The best run without a lens yielded an FID score of 52. Adding the lens helped the system stabilize and produce meaningful results in all runs, with the best run yielding FID scores of 32 and the worst run yielding an FID of 37. Note that these numbers are comparably high due to the small model size of the generator and discriminator. The effects of the lens during training are shown in the Appendix, Fig. 8.

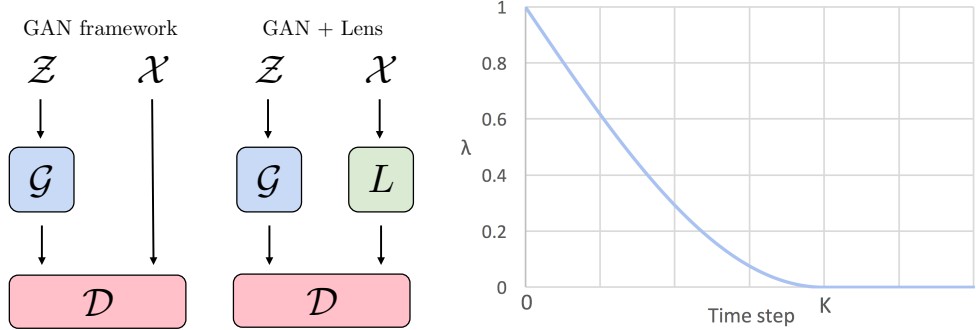

Figure 2: **Schematic of the proposed module (left).** We add a lens $L$ in between the real data distribution $\mathcal{X}$ and the discriminator $D$. The lens is compatible with any type of GAN and dataset type. It finds a balance between fooling the discriminator and a reconstruction loss, leading to a tempered training procedure that self-adjusts to the capabilities of the current generator *w.r.t.* the current discriminator. **Schedule for the weight $\lambda$ for the adversarial loss term $\mathcal{L}_L^A$ of the lens during training (right).** As training progresses, the value is lowered in a smooth way from 1 to 0 in $K$ steps, increasing the relative weight of the reconstruction loss for the lens. We set $K$=10k in all experiments. While lower values showed faster convergence rates in our experiments, we opted for a single value in all experiments for simplicity and to avoid adding yet another hyperparameter that needs to be tuned. We found that the performance is robust against changed for the specific value for $K$ and a single value to yield good results across datasets and GAN architectures.

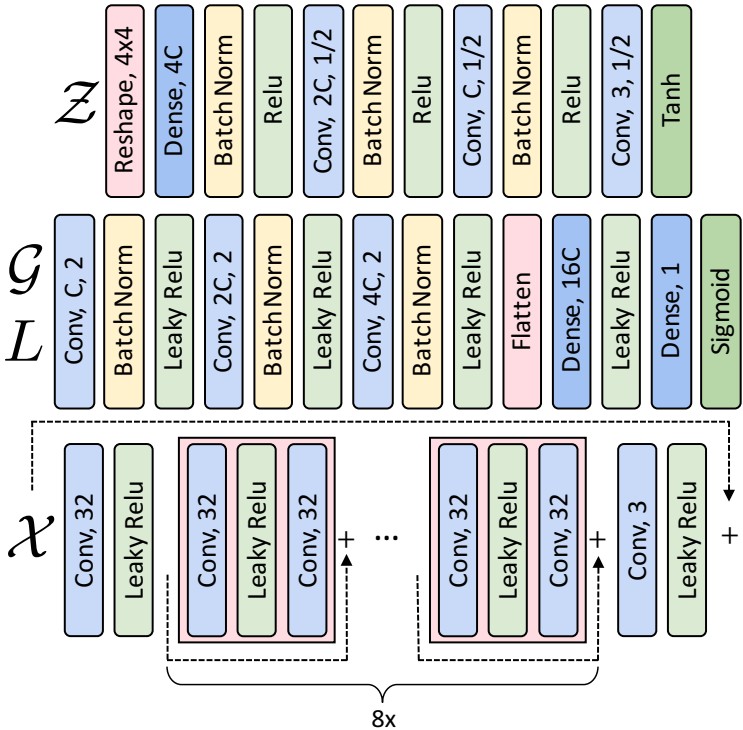

Figure 3: **Network architecture of the generator $G$ (top) and discriminator $D$ (middle).** The design follows Radford et al. (2016). The strides of the convolutions are 1/2 for upsampling in $G$ and 2 for downsampling in $D$. The kernel size is 4×4 in both networks. The number of parameters can be varied by adjusting $C$. **Network architecture of the lens (bottom).** The core of the network is composed of 8 residual blocks. To help convergence to identity, we add an additional residual connection from the input to the output. All convolutions have 3×3 kernels and stride 1.

Figure 4: MNIST digits produced by DCGAN with a lens with **fixed** $\lambda = 1$ (top). The columns show generated and lensed samples. The lens $L$ adds pertubations that make the real data samples look more similar to fake samples. As training progresses, the quality increases and the reconstruction of $L$ improves steadily. Ideally, the system would converge to a point where $G$ produces samples that are indistinguishable from $\mathcal{X}$ for a fully trained discriminator – at this point, $L$ would turn into the identity mapping. While training with the lens is very stable and while the FID was still decreasing when we stopped training, the reconstructions are not perfect even after 10M training steps and the FID is still only 60, *i.e.*, it has not yet even reached the performance of DCGAN without a lens after only 1M steps (bottom right, green curve). When the value for $\lambda$ is adapted (see Sec. A.3), training is greatly sped up and, the quality of the samples is substantially higher (FID 22) than for the GAN without a lens (FID 42). The difference is also visible in the results, where the GAN with a lens produces better looking MNIST digits. Note that the FID is initially higher for the GAN with a lens in the bottom right. This is because the FID is always measured against the real samples $\mathcal{X}$, while $G$ is initially trained for the lensed distribution $L(\mathcal{X})$ that differs from $\mathcal{X}$ in the early training stages.

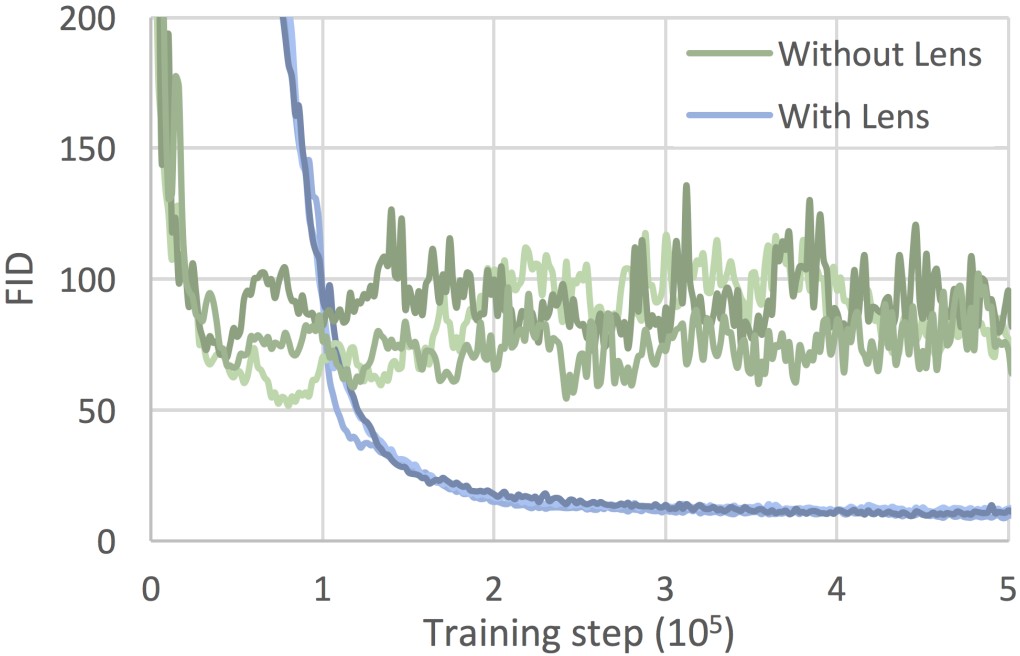

Figure 5: FID for DCGAN trained on the Color MNIST dataset. For each method, 3 independent runs with different random seeds for the weight initialization are shown. Since the value of $\lambda$ is high early on during training, the GAN with a lens initially performs worse, but the quality soon catches up and surpasses that of the GAN without a lens. The GANs with a lens are much more stable and more robust *w.r.t.* different initializations.

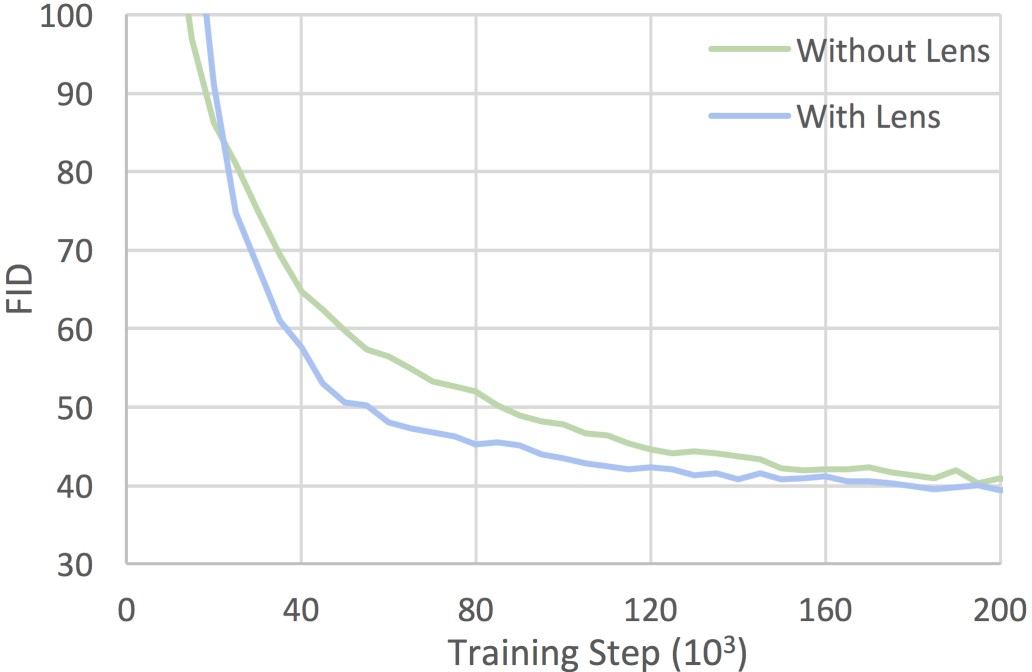

Figure 6: FID for WGAN-GP on Cifar-10 with and without a lens. The value for $\lambda$ is smoothly lowered from 1 to 0 in the first $K$=10K steps. The final results have similar FIDs, but WGAN-GP with a lens converges faster to higher-quality samples. The GAN with a lens converged faster in a number of independent runs, though only one run is shown here. Tuning the rate at which $\lambda$ is adapted could further improve convergence speeds.

Generated samples $G(\mathcal{Z})$          Lensed samples $L(\mathcal{X})$

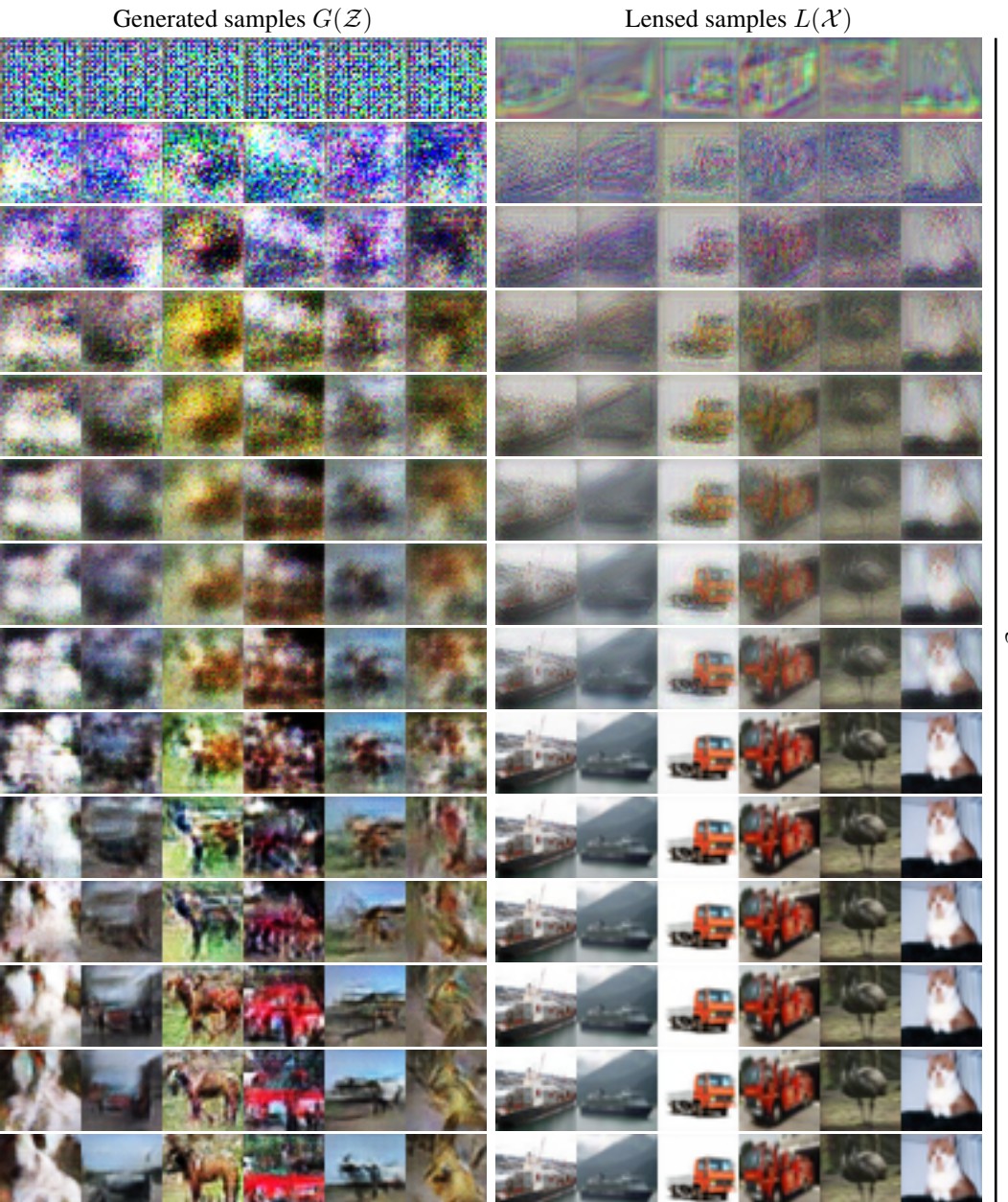

Training time

Figure 7: WGAN-GP with a lens $L$. In early training stages, the images are blurry lack contrast, but $L$ gradually reconstructs finer details as $G$ catches up. Note that by design, $L$ could easily converge to the perfect identity mapping very quickly, so the gradual improvements seen here are a result of the adversarial loss term $\mathcal{L}_L^A$ rather than slow convergence. See Fig. 6 for a plot of the FID of the GAN with and without a lens.

Generated samples $G(\mathcal{Z})$    lensed samples $L(\mathcal{X})$

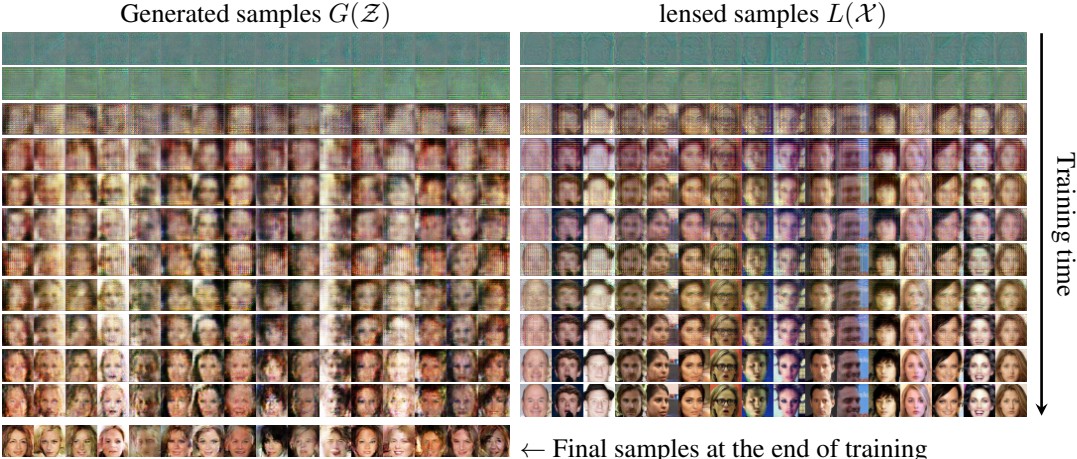

← Final samples at the end of training

Figure 8: Generated and lensed samples at various steps during the training process of LSGAN on the CelebA dataset with a lens. The generator produces a large variety of faces since it is not forced to reproduce fine details early during training, making it less prone to the mode collapse problem.

