# OpenReview forum: "Tempered Adversarial Networks"
_ICLR.cc/2018/Workshop — Accept_

### Official Review · AnonReviewer3 · 2018-03-09
**Good workshop paper**

**Rating:** 7
**Confidence:** 2

**Review:**

This paper proposes an interesting idea that puts an additional "lense" layer before the real training samples, to cope with the initial big difference in distribution between generated and real samples. The lense layer is gradually tuned towards identity over training. Quite thorough experiments show the idea is kind of working and indeed make the training stable. So I think it could be a good workshop paper.

I wonder how it is compared to BEGAN (https://arxiv.org/abs/1703.10717) which also uses a hyper parameter to tune the strength between Generative and Discriminative components.

Note that I am not an expert in GAN so I might miss some strongly related works.

---

### Official Review · AnonReviewer2 · 2018-03-09
**Interesting, but not convinced by the setup.**

**Rating:** 6
**Confidence:** 4

**Review:**

In this paper, the authors propose to add a "Lens" network between the true data distribution and  the discriminator. They train this network with two objectives, the normal discriminator objective L_D, and a reconstruction objective.
The authors then report improved results on multiple datasets and architectures.

The paper is well-written from an essay point of view, but from an experimental point of view, several things are fishy and worrying. The story that the authors propose doesn't satisfy me, here's why:
1) in GANs, generators do *not* chase a fixed distribution. They chase the implicit data distribution that the discriminator has modeled through gradients provided by said discriminator.
2) the proposed story for the Lens mechanism is that it "automatically balances" L^R and L^A, but there is little evidence to suggest that this is the case. In fact the evidence, to me, suggests that:
2.1) as L^A starts going away things get better (given that you choose K=1e5 and that from your training curves things start going well at that point, I'm wondering why you have no training curve for lambda=0; Fig 4's explanation is also conjecture IMO)
2.2) as L^R gets lower because of a better reconstruction, things get better.
3) Once lambda is removed, there is some resemblance with "Adversarial Autoencoders" (arxiv.org/abs/1511.05644), and with "Adversarially Learned Inference" (arxiv.org/abs/1606.00704). (I'm not on author on either those papers)
4) You claim that the effect of the lens is "automatically" balanced, but you've very clearly defined manually a scheduling for lambda, and it is not obvious to me why L minimizing L^A balances anything.

w.r.t. 2.1, what would happen if you froze everything, reset L(X), and then train only L(X)? Would it match G(X)? Would it match X? (as a result of minimizing L^R).
What would happen if L(X) simply optimized for L^R? Maybe that is the real contribution here.


There's definitely something going on because you are getting interesting results, but I think your setup deserves a lot more investigating, just for example:
- The discriminator's data is coming from a neural net, that's somewhat unusual. Does that have a meaningful effect?
- The lens network could cheat with the residual connection, does it? If not is it really helping out the discriminator? How?

While I'm aware this is a workshop submission and not all those questions can be answered, the authors seem to be selling more than they have which bothers me. They make a (afaik) novel contribution, have results to back what they're doing.

Side note: It says on the iclr.cc website that "Extended abstracts submitted to the Workshop Track are strictly limited to 3 pages, excluding references." To me this paper seems like a full paper disguised as an extended abstract with a long appendix.

Typo: in Section 2 'G(\mathcal{X})' should be 'G(\mathcal{Z})' (twice).

---

### Official Review · AnonReviewer1 · 2018-03-09
**the advantages of the proposed methods are not clear**

**Rating:** 5
**Confidence:** 5

**Review:**

summary:

The paper propose to learn a function L that is almost the identity (min _{L} ||x- L(x)||) for training GAN where (real is replaced by L(x)), L is progressively going to identity. This is similar to the effect of adding noise that is annealed over the training on real data.

Asymmetric noise on real data only  has been considered for  in Sobolev GAN. Noising both real and fake data was also considered known as instance noise in "AMORTISED MAP INFERENCE FOR IMAGE SUPER-RESOLUTION".  A comparaison to annealed noising should be considered in this paper,as L is only learning some noising/ blurring of the image.

The paper mentions some relation to progressive growing  and says that the scale can be only used for images, the paper will be much stronger if it shows any improvement using the lens on text generation for instance.

Overall the ideas in the paper need more empirical investigation: 1) comparaison to noising and 2) seeing if the lens function gives improvement in text generation where progressive scale growing is not obvious.

---

### Author Response · Authors · 2018-03-03
**Erroneous figures**

We have noticed an error in the figures of the submission. Several figures in the uploaded version erroneously show generated samples instead of real or lensed real samples due to a bug in a post processing image cropping script for the tex file.

Details:
Fig. 1+4: top right column "Lensed Samples L(X)" and "Real samples X" below mistakenly display the results of the Generator.
Fig. 8: top right column "Lensed Samples L(X)" mistakenly shows the same images as the generated images.

The figures have been updated with the correct images in the arXiv version (1802.04374).

---

> ### Public Comment · ~Keeran_Adams1 · 2022-03-28
> **Tips for help**
>
> Thanks for the information! This turned out to be very helpful! You can also ask https://www.edugeeksclub.com/pay-for-essay/ for help with essay writing. Quite thorough experiments show the idea is kind of working and indeed make the training stable. So I think it could be a good workshop paper.

---

### Author Response · Authors · 2018-04-15
**Thank you**

We thank the authors for the helpful comments and will incorporate the suggestions into ongoing efforts to further analyze the effects of the proposed technique.

---

### Decision · Program_Chairs · 2018-03-20
**ICLR 2018 Workshop Acceptance Decision**

**Decision:**

Accept

**Comment:**

Congratulations, your paper was accepted to the ICLR workshop.